# 'Un-Central' Landscapes of NE-Africa and W-Asia—Landscape Archaeology as a Tool for Socio-Economic History in Arid Landscapes

### Anna-Katharina Rieger

Institute of Ancient History and Classical Antiquities, University of Graz, A-8010 Graz, Austria; anna.rieger@uni-graz.at; Tel.: +43-316-380-2391

**Abstract:** Arid regions in the Old World Dry Belt are assumed to be marginal regions, not only in ecological terms, but also economically and socially. Such views in geography, archaeology, and sociology are—despite the real limits of living in arid landscapes—partly influenced by derivates of Central Place Theory as developed for European medieval city-based economies. For other historical time periods and regions, this narrative inhibited socio-economic research with data-based and non-biased approaches. This paper aims, in two arid Graeco-Roman landscapes, to show how far approaches from landscape archaeology and social network analysis combined with the "small world phenomenon" can help to overcome a dichotomic view on core places and their areas, and understand settlement patterns and economic practices in a nuanced way. With Hauran in Southern Syria and Marmarica in NW-Egypt, I revise the concept of marginality, and look for qualitatively and spatially defined relationships between settlements, for both resource management and social organization. This 'un-central' perspective on arid landscapes provides insights on how arid regions functioned economically and socially due to a particular spatial concept and connection with their (scarce) resources, mainly water.

**Keywords:** aridity; marginality; landscape archaeology; Marmarica (NW-Egypt); Hauran (Syria/ Jordan); Graeco-Roman period; spatial scales in networks; network relationship qualities; interaction; resource management

## 1. Landscape Archaeology and Central Place Theory

Central place theory (CPT)— developed by Christaller 1933 [1]—revolves around human agents, settlements, and economies. It is about location, connections, and hierarchies. Apart from many useful applications of this theory as a model for explaining and generalizing patterns of settlements, centralized services, and flows of goods [2], the main point of criticism is the particularity of the historical situation from which it was developed: European medieval cities as centers of production and consumption, trading systems and territories. Central place theory has been and can be adapted to areas and historical settings other than medieval Europe [3], since its evaluations are based mainly on economical parameters. However, with this rather limited spectrum of parameters, it dominated the historical-archaeological thinking of European and American academia for a long time. As such, CPT was applied to modern regions and economies [4,5] as well as to many places and regions of Classical Mediterranean antiquity, perceived as times and regions where social and economic life was organized by and in cities [6,7]. Cities are a major feature of social and spatial organization in Mediterranean archaeology. Yet, cities are often still a major focus of research due to the density and concentration of material evidence, which is the ease of access to archaeological and textual

sources focused on one topographical spot. For these reasons, CPT has been a long-serving guideline in archaeology.

CPT was developed from and has been applied to agriculturally exploited regions with cities being complex places where people lived. Recent adaptations of CPT start from this fact and question the concept of territory, the definition of city, or the dichotomy of center and periphery [8,9]. Terms like 'central' and 'centrality' can only be applied if a hierarchical organization of society and hierarchical distribution of power and resources can be claimed. Whether something is central depends first on the point of view and then on the spatial scale of the research [9]. When studying past societies, where archaeologists and historians use material and/or textual sources, the resolution of the extant source material for the analysis (a route network, a product or commodities, names, or descriptions in texts) determines the resulting centrality. Centrality is thus a relative term.

With the rise of landscape archaeology in the 1970s [10–12], a shift in how to relate settlement patterns, material culture, and landscapes occurred. Landscapes started to be perceived as large-scale places in which spatial practices and networks are at work, as is also the case in urban environments [13]. Landscapes can be understood as a palimpsest of history, human agents, economy, technology, culture, memory, and almost all human expressions [14,15]. Landscape archaeology, at its core, is area-oriented and not site-oriented, but note the significance of 'places' for agents' practices in and with landscapes [16,17] [18] (p. 44). What might appear as a central place, a settlement, facility, city, source, or the like, is not investigated first, nor is it analyzed in the larger context of socio-spatial practices preserved in the landscape. An advantage lies in the fact that more find spots are integrated into a socio-economic and socio-religious history, and different findings are used and analyzed at the same heuristic level. In practical landscape-archaeological work, off-site findings are considered, but off-site still implies a site from which the of'-site is distant. Here again we risk applying hierarchical order to the analytical process. As is the case for central places, we should look for a more balanced terminology for not repeating pre-determined categorizations that can bias our interpretations [19] (pp. 5–8 for criticism).

This view on landscapes opens new processes for considering parameters, like interaction and relationships, to better understand inter-human interaction, economic systems, trade relationships, habitational traditions, or centers. The combination of the approaches of central place theory and landscape archaeology is then a promising method for examining the past in a dynamic and not a hierarchical mode, which is still common in European academia even after the attainment of post-colonialism.

Against this theoretical and methodological backdrop, I present two case studies that may demonstrate the efficacy of the combination of landscape archaeological approaches reflecting on centrality. The paper is based on a comparison of datasets from two ancient arid regions: the Hauran in Southern Syria and Northern Jordan, and the Eastern Marmarica in NW-Egypt in Graeco-Roman times (second and first century BCE to third and fourth century CE), which I compiled and studied in completely differing research contexts and with differing approaches. I analyzed how centrality thinking and how teleological interests in being at the center (or being close to the center) influenced our historical reasoning. By describing the phenomena of the two areas, I lay the background for the approach and analyze the results from an 'un-central' viewpoint on landscapes.

With help from network analysis, landscape archaeology and a revised CPT were successfully combined. Degrees of connectedness; parameters for centrality; and the soft (assumedly subjective) factors influencing how and in what respect to a place, a region, or landscape is important reflect the ongoing difficulties of humanities in working with hard (assumedly objective) and soft (assumedly subjective) factors contemporaneously, contrary to the natural sciences [20–22] (the latter two chose an agent-based approach). Recent studies on emotional decision-making in the economy [23,24] show the human influence on every realm of economic, financial, and (also in modern times) medical interactions. It is not only a matter of ideology and power relationships, but also about barely

measurable advantages or influences on how people behave, identify, or react (for ancient economic systems [25,26].

## 2. Thinking 'Un-Centrally'—Where to Start and What to Ask

The two case studies, I offer options for rethinking whether centrality has automatically indicated powerful and dominant. The Eastern Marmarica, on the fringes of the Libyan Desert, as well as the Hauran west of the Arabian Desert (Figure 1) are arid landscapes in the Old World Dry Belt, and only partly suitable for agriculture. Hence, the land was not only in ecologically but also socio-economically marginal. To choose marginal, arid areas for questioning the focus on central places in archaeology and other disciplines shifts our focus to the assumedly non-important and non-central. The outcome of this shifted focus may explain the significance of thinking 'un-centrally'.

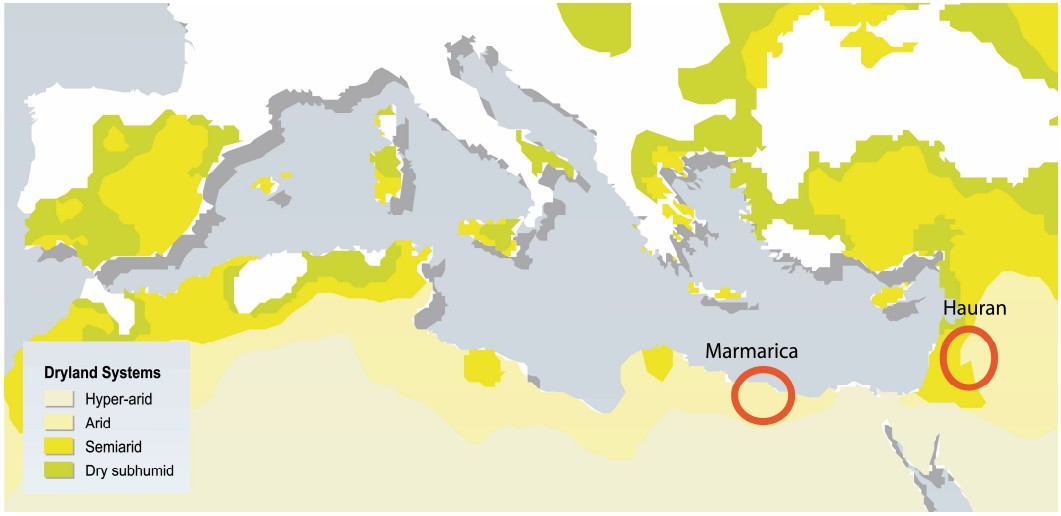

**Figure 1.** The Mediterranean and the MENA-region (Middle East and Northern Africa) showing the grades of aridity. The red circles mark the areas of the two case studies: the Marmarica in NW-Egypt (left circle), the Hauran in Southern Syria/Northern Jordan (right circle). Both are semi-arid to arid landscapes. Base map: [27] (p. 23).

The data for the Eastern Marmarica (Section 5.2) are the product of landscape-archaeological fieldwork, where the main focus was on the interaction of mobile and sedentary life-strategies, which are dependent on water availability. Since only few textual sources refer chiefly to the region, the main hermeneutical methods were archaeological, geodetic, and geo-hydrological surveys. The chronology was established through pottery analogies of the local production; the results for the ancient water management were based on an integrated analysis the relief, the soil, and calculations.

The study on Hauran (Section 5.1) was developed as part of the framework of a project on the history and archaeology of religion (see Acknowledgements) that focused on the socio-spatial organization of sacred spaces. The data do not relate to my own fieldwork but to published data from French and German projects on the Graeco-Roman phase. Here, the analysis of water is the connecting link to understanding not only individual places, but also their position in making the region function through the water distribution. Hence, accounting for the water scarcity in the two arid regions allowed for the recognition of the logic of their socio-economic organization.

Un-central thinking can be applied to various socio-spatial organizations, such as ecologically extreme habitats like mountain areas, rural areas, or egalitarian groups. In my case studies, the ecological extreme of aridity is the common strand.

Aridity can be defined as follows:

- Resources: limited availability of the resource water and soil
- Climate: high temperatures with high day-night variability, high evaporation rates

- Vegetation: steppe to concentrated to no vegetation
- Population: sparsely populated areas
- Economy: limited economical potential

As a result, the life strategies of the inhabitants of arid regions revolves around social (and religious) organization and institutions, revolving around resource management and a combination of different lifestyles.

Marginality is not only used for ecological phenomena but also as a descriptive term for socio-cultural relationships or positions. The issue of marginality is discussed below in more detail. For an initial overview, the following points characterize marginality:

- Resources: limited availability of or access to resources
- Life strategies: spatial and habitational (for humans social) and temporal niches
- Population: little access and/or participation in larger networks, political power, and cultural institutions
- Economy: limited economical potential

Since centrality, as developed by Christaller [1], starts from economic potential, reviewing it in the scope of this special issue on "un-central landscapes" starts from zones that are not normally regarded as economically powerful, that is, ecologically and economically marginal. Thinking un-centrally embraces a methodology that does not focus on the site or the place, but on the area, the spaces, and—in particular—the relationships, connections, and mutual relationships into which people, objects, and natural phenomena are embedded. Un-central thinking that is influenced by network analysis and actor-network-theory, however, also remains close to the problem of concentration in the network analysis on hubs and nodes (see below). However, the socio-spatial patterns and socio-economic patterns of past groups and societies are better elucidated if we apply a less site-oriented view. Thinking of relationships is also part of central place theory: routes, connections, contacts, and exchanges are thematized in the parameters of how to define centrality (see the prudently developed enhancement of CPT [28,29]). The problem partly originates from the fact that central place theory is a city-based theory as are many socio-spatial studies and researchers [30] base their arguments on urban contexts. What is not considered are the qualities and intensity of relationships, which also implies a temporally different intensity or existence of relationships. The same is true for the various centralities as defined in network analysis: degree and betweenness centrality do not specify all qualities of the centrality. Degree centrality depicts the number of neighbors and links that a certain point or node has to them; betweenness centrality shows those important points or nodes through which shortest routes pass. Yet, the definition of central is defined by presumptions ('neighbor' or 'short') that do not reflect all the possibilities of human perception, experience, and decision-making.

Too often interpretations of the relationships between objects, environment, and people, which we assume to see in the archaeological record, fall short, since, in the wake of the teleological construction of history and the progressive development of mankind and history, we tend to look for hierarchical relationships. Too rarely do we consider correlations that are the basis for the relationships, qualities, intensities, and their changes [28,30]. To set out my agenda briefly, I: (1) differentiate the view that un-central and marginal places and areas are generally equated with the non-powerful and low economic potential; (2) offer an example in how the close look at the dependencies and relationships of landscapes and people revises the view on marginal areas and places; and (3) deduce my conclusions from the social, religious, and economic practice of resource management, in my cases, of water management

## 3. Marginality Reconsidered

### 3.1. Marginality—A Concept to be Differentiated

Marginality can be treated as a social or physical-geographical phenomenon. Social marginality or marginalization means that groups or individuals can be excluded from access to political power,

cultural or economic resources, or housing, etc. [31] (p. 7). With the last example, housing, a spatial parameter is influential: spatial or geographical marginality is, in many cases, related to social marginality (or vice versa) [32]. Research in human geography bridges the gap between social and spatial marginality [33] (p. 90) [34,35]. However, most studies start by looking at urban contexts, including cities or urbanized areas. Even more specifically, they are interested in modern cities and economic systems. Agricultural areas and regions that are not structured by cities and other focal places were not and are not the focus of research [35,36]. This is true also for Graeco-Roman antiquity. Marginal societal groups came to the fore in studies on children, women, slaves, or marginal professions. Spatial aspects of marginality are not an issue in research on ancient Mediterranean societies and their habitats. The importance and value of marginal habitats, spaces, and socio-economic practices for a historical understanding are rather accounted for in Pre- and Early History [28] (p. 2) [37,38].

Another constraint is the criteria for socio-political, economic, and ecological marginality not being mutually exclusive, as Young and Simmonds [38] already criticized. One criterion can be at work, yet the others do not have to be applicable. Moreover, the dichotomic view—paralleled with center and periphery—preconditions hierarchical orders and has been proven to be too simplistic and static. This is also true for the predominant narratives about people at the margins as being disadvantaged and backward, living with an assumed permanent experience of being marginalized. These might be (or have been) often close to reality. However, differentiations and non-dichotomic, non-biased analyses can help with comprehending the social, economic, and spatial practices of different and multifarious past and modern societies or groups.

The geographical and ecological marginality of past societies and regions received more attention in the wake of landscape archaeology [39,40]. A recent study [37] provides a comprehensive overview on the problematic definitions of marginality but proposes overly rigid scales for measuring the marginality of regions or habitats in the four main fields of social, ecological, economic, and spatial marginality. Firstly, scales of measurement as such are difficult to apply to a factor like 'social marginality'. How can we measure the intensity and impact of social marginality? Is it an emic or an etic view we apply? Secondly, the normal lack of information in archaeologically studied societies and groups leads to sets of non-comparable data [37] (e.g., Figure 3). Knitter and Nakoinz [28] generally pursued a convincing approach. Yet, they measured "spheres of influence" of certain aspects of a center (administration, trade, craft, etc.) in spatial distances to measure the "intensity-level of centrality" [28] (p. 5, and e.g., Figure 4). Even though the addressing of socio-cultural phenomena, like trade, by mathematically based least-cost-paths or the computer-aided reconstruction of networks by degrees and categories of calculable degrees and qualities of centralities is a method for understanding the past, their risk lies in the rigidness of the formulae.

The areas of socio-economic and socio-spatial practices have to be described when we study marginal areas or groups in archaeology. However, the focus should lie on the interdependencies and the relationships, and their various intensities or temporal actualizations. How dispersed or concentrated, how remote or close the settlements, supply areas, routes, or persons are already biased descriptions: distance or closeness, or accessibility or means of communication are socially constructed practices. Whether one is economically successful or fails is also dependent on viewpoint. Whether the concentration of functions, services, and facilities in a center is perceived by everyone as such or is permanently virulent depends on the standpoint of the visitor, inhabitant, or other agents.

Hence, only adapted approaches on marginality and centrality provide the "relative concentration of interaction" [28] (p. 4; with a feasible differentiation between "centrality potential" and "actual centrality"), where marginality can be conceptualized as the "relative absence of interaction". The case studies below demonstrate on a certain spatial scale how land-use and habitational patterns in ecologically marginal regions can be conceptualized differently when seen through the lens of interaction, and how marginality applies only to limited areas in the lives of the population.

### 3.2. Weak and Strong Ties and the Study of Marginal Areas

In searching for interactions, relationships, and connections, historians and archaeologists use approaches from network analysis [41,42]. The much more dynamic picture derived from network graphs than from nested geometric graphs of CPT accounts for the more dynamic view of inter-human-landscape interactions [37,43,44], upon which landscape archaeology is mainly concentrated. The less hierarchical view practiced in network analysis focuses on the interdependencies instead of monodirectional pathways. However, when archaeologists and historians study landscapes, trade connections, settlements, and land use as patterns, as well as the distribution patterns of objects, they tend to investigate the nodes and hubs in the network instead of the edges [44]. They do not consider the factors, phenomena, or impacts by which the edges are, or can be, formed and influenced [45,46] (p. 170 with n. 5, for scaled approaches). Is it a physically existing road that connects people and places or is it rather the people using this road, that we look at as edges? In the latter case, the edge is human and mobile and not a physically traceable track. What objects traveled and what ideas traveled with it [47]? What if the connection is only mental, for example to a goddess or to a notion, as I will describe in the case of water and religious institutions?

Hence, all triangulations, cost-path calculations, and categorizations of centrality or marginality, of betweenness or degree centrality, depend on (assumedly) soft factors. This does not mean that the concepts are not useful and actually often lead to intriguing results [28,42,43]. However, history, archaeology, and social sciences revolve around human beings and their position and role in the world, with all human constitutions and conditions. These gaps between (assumedly) hard and soft factors have to be conceptualized and bridged by methodologies including the (assumedly) soft factors. The laws of diffusions and rules for concentrations of objects, people, and ideas, as well as measures for connections and relationships between them at certain spots or in certain areas, are subject not only to measurable parameters [45,47]. Here, the differentiation of how marginality can be perceived and the recourse to Granovetter's model of weak and strong ties meet [48,49]. Strong and weak ties introduced to the network analyses enlighten the not-always-straightforward or expected methods of diffusion and concentration. Even though he studied social ties and the diffusion of a certain kind of knowledge (rumors, for example, spreading with people), his findings can be generalized to other realms. He argued convincingly for the emphasis of weak ties as constituting relationships in and among societies or groups. People who are spatially and socially not close to each other are the nodes in a network through which the largest spread or new concentrations of diffused knowledge (or objects persons, ideas) can be reached. Weak ties can bridge large social (or spatial) gaps because they are bridges and not fully-fledged relationships [48] (p. 1364–1366, Figure 2). The value of this shift of perspectives is that the segments and margins of the groups that are emphasized. People at the margins have the position and capacity to connect (distant) groups. It is not the people (and places) well-embedded with many contacts creating networks; it happens through the rather isolated people or hubs. This applies especially to the case study of Marmarica.

Applied to a more spatial approach, the weak and strong ties correspond to bridges, short cuts, or longer and shorter distances, but also to intensity that does not regard distance or closeness. Acknowledging weak and strong ties defines the quality of the relationships, or the randomness of the (social) ties, allowing for the dynamic organization of connections, but also of detachments. The so-called "small-world phenomenon" [50] combines the (social and spatial) distance and the quality of connections, and accounts for either randomness or clusters in networks [36,51] (for an application to the Mediterranean past pp. 27–33).

The un-central, less biased view in combination with the differentiation of the weak and the strong ties help develop a new perspective on marginal areas, as well as marginalized spaces and people. The organization of relationships, as well as the spatial and temporal scale on which the organization is practiced, play a role in the model of weak and strong ties between the agents and define the "small" or "large" worlds. An application to a geographical region and a social organization in a certain period of time leads to new and differing views of settlement patterns and socio-spatial and socio-economic

organization. In the Graeco-Roman Hauran, these were based on strong ties, and led to a rhizomatic network of places and settlements related to each other by water management. In the Graeco-Roman Eastern Marmarica, rather weak ties and bridges were at work and led to a tree-like dependence on settlements and places.

## 4. Marmarica and Hauran in Graeco-Roman Times: Two Arid Regions—But Are They Marginal Regions?

For easier communication, we work with definitions, terminologies, and categories (arid, marginal, Graeco-Roman, and central). This is the top-down method of communication. However, what difficulties or opportunities do we face when we proceed bottom-up, starting from a physical-geographical, ecological, historical, and archaeological context? With the two regions in the scope of this paper investigated mainly in their Graeco-Roman period, I can exemplify the impact and relevance of soft (assumedly subjective) factors on the definition of marginal or central. The two arid regions act as cases for interpretational openness and methodological accuracy that allow for re-narrating histories, or at least for adding new aspects to these histories [52,53] (p. 85 and 96).

Hauran in Southern Syria/Northern Jordan, and Marmarica in Northwestern Egypt are both arid environments (Figures 1–3). The two regions allow for a vision (Marmarica) and revision (Hauran) of repeated and perpetuated views on desert, steppe, or marginal areas from the viewpoint of the Mediterranean and Graeco-Roman spheres of influence. These views lead to interpretations of socio-cultural settings and phenomena in the sense of provinciality, closeness (loyalty) to, or remoteness from (aversion to) Roman rule [53–56]. Central place theory might be seen as a continuum in such interpretative patterns by (western) archaeologists and historians. However, starting from the region, its historical and ecological conditions and socio-economic organization led to differentiated views. Not focusing on central places, and only sites in the archaeological sense, allows for a more dynamic view of interhuman and human-landscape interaction. The landscape of a region is not limited to the exploitable hinterland. The inhabitants rely on it, as well as they form and are formed, adapt and adapt to the landscape as their place of living, memories, death, and notions of the divine (cf. e.g., [57]).

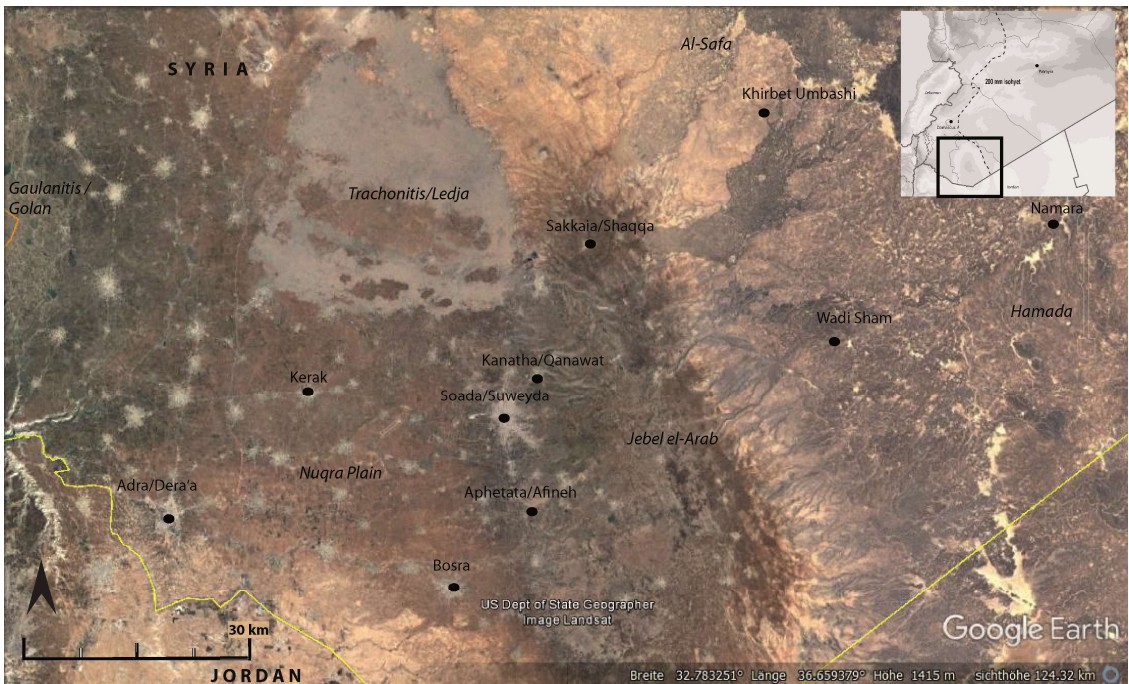

**Figure 2.** Satellite image of the northern parts of the arid landscape of Hauran (Southern Syria) showing geographical units and settlements referred to in the text. Created with Google Earth V 7.3.2.5495 (Google, Mountain View, CA, USA). Image: Landsat/Copernicus 2016, as base map.

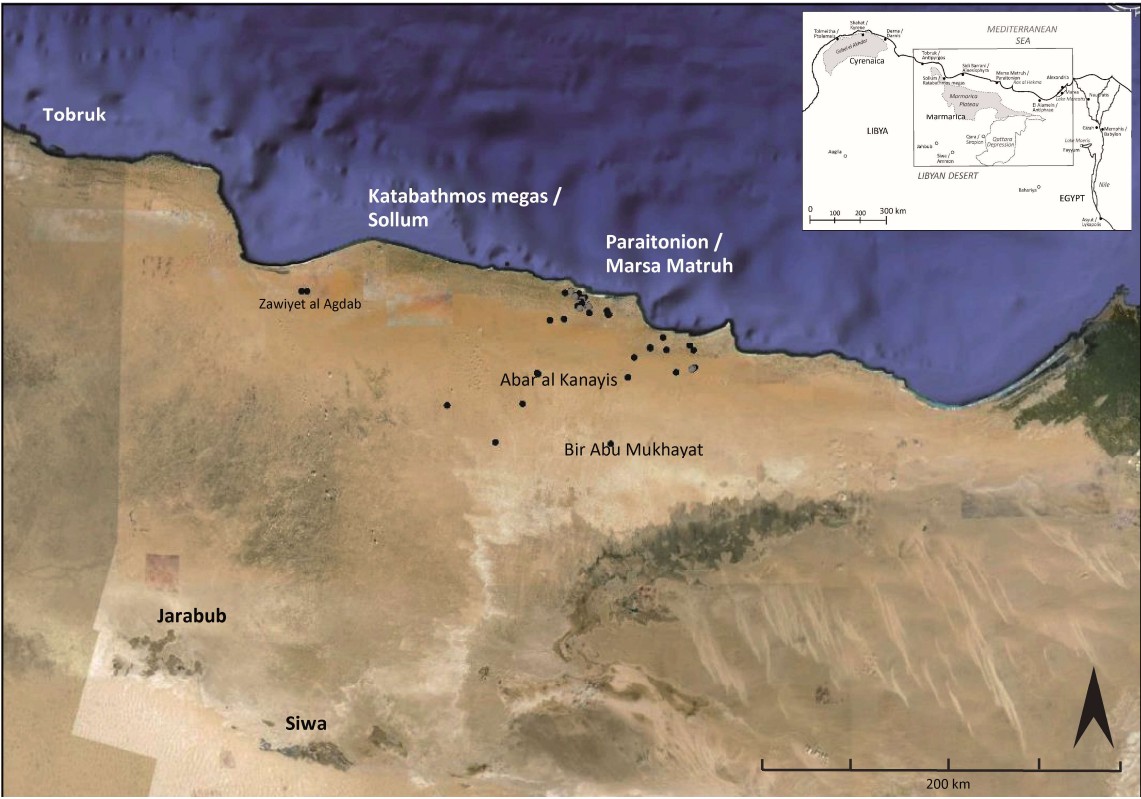

**Figure 3.** Satellite image of the eastern parts of the arid landscape of the Marmarica showing geographical units and settlements referred to in the text. Created with Google Earth V 7.3.2.5495 (Google, Mountain View, CA, USA). Image: Landsat/Copernicus 2016, as base map.

The Hauran, with its sub-units called Gaulanitis, Hauranitis, Batanea, and Trachonitis in antiquity, is a mainly basaltic region situated in the Yarmuk drainage basin [58] (see their Figure 6), and characterized by variable precipitations where only Jebel el-Arab (1800 m a.s.l.) receives enough rain for rain-fed agriculture (Figure 2) [59,60] ([59], their Figure 1 shows the 250 mm isohyets crossing the region). To the east the Safa and the Hamad, parts of the Arabian Desert adjoin.

The Marmarica is situated on the northern fringes of the Libyan Desert (Sahara), between the Nile Valley and Cyrenaica, as well as between the Mediterranean coast and the Qattara Depression/Siwa Oasis. In the calcareous region, precipitation is insufficient for rain-fed agriculture and steppe vegetation dominates (Figure 3) [61,62].

Water was and is the limiting factor for economic activities. Wide-spanning systems of water provision, water-collection, and water-distribution, typical for arid areas, characterize infrastructure, social institutions, and the economy [61–63]. Given the variable resource availability, livelihoods are based on a mixed system of nomadic and sedentary life strategies. Hence, the varied economic potential of the marginal areas is exploited or utilized: a kind of opportunistic agriculture, combined with livestock breeding, and participation in and organization of long-distance and short-range trade and exchange of goods [62].

The population in Marmarica has lived there since the Late Bronze Age up to present, partly sedentarily, partly nomadically. However, according to the findings, a peak in agricultural production in Graeco-Roman and Byzantine times allowed a surplus economy [64–66]. Hence, one marker for marginality, the economic potential of a region, is affected by human impact. The impact is the extensively mastered and managed water harvesting installations along, and in the wadi as well as on the tableland of the Marmarica Plateau. For a period of ca. 500 years, the area was far from economically marginal.

In Hauran, traces of human presence start in Neolithic times when the region was already as dry as in later times apart from minor shifts (e.g., during the Roman Warm Period) [67–70]. The earliest traces of many settlements date back to the first centuries of the first millennium BCE, during which many underwent a heavy reconstruction and enlargement in Graeco-Roman times. Despite the harsh conditions in the Trachonitis (Leja) or in the areas north and south of Jebel el-Arab, the settlement density was fairly high.

In terms of political and administrative history, the two areas reflect their position on the fringes. In Marmarica, despite the attempts of the Egyptians from the Nile Valley, official Roman rule began in 64 BCE, when it became part of the province of Creta and Cyrenaica. In the eastern parts, this happened earlier: with the defeat of Cleopatra in 31 BCE, the formerly Ptolemaic Eastern Marmarica formally came under Roman administration during a political re-organization of the region rather than during an economic peak [71,72].

In the case of Hauran, the Romans took over rule in 64 BCE in the northwestern part (Syria) and in 106 CE in the southeastern parts (Arabia). Earlier, the Ptolemaic and Seleucid kingdoms influenced the region, which was politically organized into city associations (Dekapolis). The areas of the Gaulanitis (Golan), the Mount Hermon, or the Safa were less urbanized, and for some periods of time, also less controlled [73] (pp. 31–53; 206–239) [74] (pp. 27–126; 412–414; 421–430).

Since both areas are on the fringes of the desert—marginal areas—they were not of much interest for the kingdoms of Ptolemies and Seleucids, since, for them, the regions also lay on their fringes. Indigenous people continued to live as they had before. However, infrastructure and population patterns changed over the decades and centuries, such as, for example, through military activities (at Mount Hermon in Hellenistic times against the Nabateans) [75] (p. 156–161) or the settling of veterans (Marmarica), as well as the need for higher crop yields due to denser populations, or economic demand (pottery production and wine production in Marmarica; grain in Hauran, see below).

As a result of the still-limited economic potential, the population density was and is not very high. Accordingly, the material culture was quantitatively not high and its categories were not multifaceted. This applies mainly to Marmarica, where settlement remains, agricultural installations and pottery constitute the main corpus of find material. Hauran, due to higher water availability, was not as sparsely populated as Marmarica, and materials ranging from inscriptions, sculptures, settlement and public architecture to agricultural installations are present.

## 5. Looking for Agents and Interactions in the Landscape on a Regional Scale

The review of marginality through the lens of central place theory, of weak and strong ties, and of interaction and relationships, can now be applied to ancient Hauran and Marmarica. The study of marginal areas should start at the local and intra-regional scale, then progress to an inter-regional scale. The Roman Empire, as a homogenizing entity, is only addressed at a higher scalar level.

### 5.1. Hauran

Hauran offers a test case for rewriting archaeologically-based history from the perspective of un-centrality. Hauran and its archaeological material, as mentioned above, academically faces a paradoxical situation: up to the present-day war in Syria, well-preserved remains of all kinds of settlement and infrastructure, sculpture, and epigraphy were found (Figure 2) (see Freyberger and Ertel [55] for epigraphy, and Meynersen [56] for sculpture). Yet, the history and archaeology of Hauran is often left in its niche of being exceptional. Hauran is often viewed only through the lens of its positions or was ascribed to between the more central powers like the Nabataeans or the Romans. How can we explain the rich corpus of sculpture, and why were they so keen on writing texts? What were people concerned about, and to whom did they feel more related or inclined to exchange with, and to whom less so? More abstractly formulated, what were the social, material, and spatial relationships, and how were objects, landscape, and people interrelated?

Studies about settlement patterns in Roman times (second to fourth century CE) focus on settlements that had the status of a *polis*, or on particularities, like *metrokomiai* [75] (p. 159 despite the balanced views, Butcher speaks about the 'striking' independency of villages) [76,77]. Why were there so many fully-fledged settlements at distances of two to four km? What were central places and what were the connections like?

The question of categorization and hierarchy is interesting to examine and revise in the case of Graeco-Roman Hauran. I researched the water management system as water is, per se, non-static, and a resource that needed redistribution in this area due to the climatic conditions described above. My research showed a strong dependency of various places in the Jebel and on the plain of Nuqra (Figure 4) [78] . The Jebel area possesses springs and receives more rainfall, whereas the plain lacks these sources of water [59,60]. Water supply is physically organized via an elaborate system of artificial and natural channels, wadi beds, source-catchments, and aqueducts (Figure 5). The water flows down to the provincial capital of Bostra, which, due to the accumulation of facilities and services, can be called a central place, from the first century (Nabataean) BCE to the fifth century CE. However, Bostra lies downstream in the plain, which, in terms of water distribution, was in the receiving position, not the providing position. In the distribution of water from natural sources, the upstream party is normally considered the stronger party [60] (Figures 5–7, here adapted in Figure 4) [68] . Even though the upstream places in the Jebel fed water into the system of water distribution, they did not automatically acquire a more powerful status. Additionally, Bostra, as a capital and powerful city, is not the first to receive water. The relationships of water distribution were based on a mutuality that was deeply rooted in social and religious institutions, of which evidence is available for the first to third century CE.

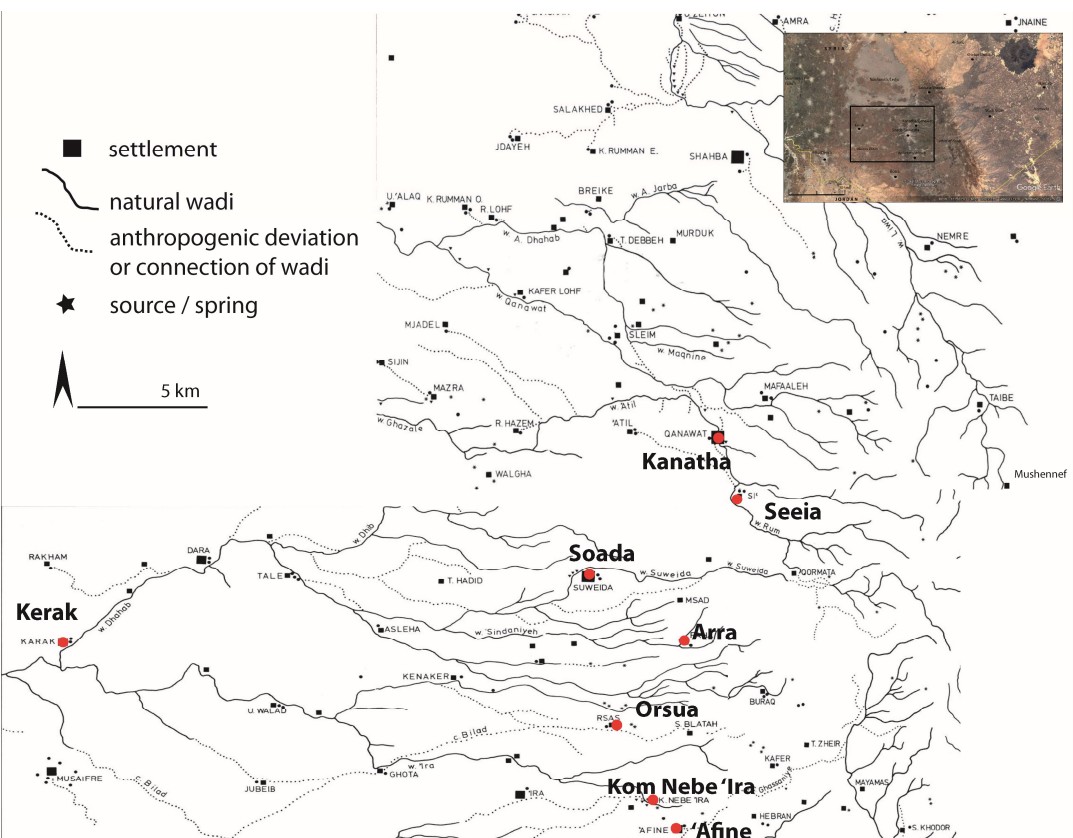

**Figure 4.** Map of wadis, sources, and deviation channels in the northern parts of Hauran. Assembled by the author from F. Braemer [60], his Figures 5–7 as base maps.

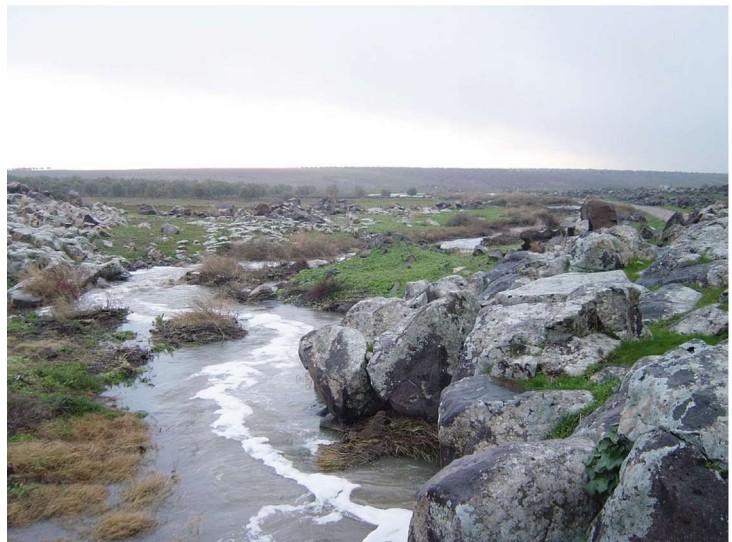
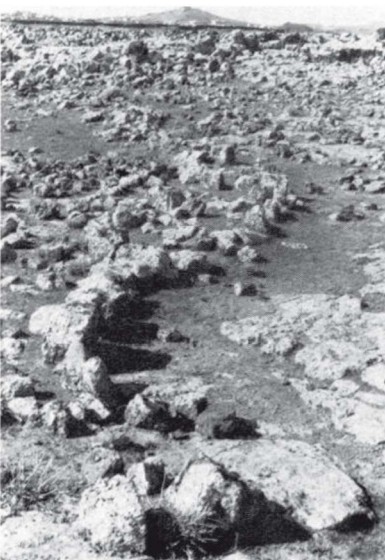

**Figure 5.** Examples of canals or conducts and wadis in Hauran. Canals are ca. 1.5 m wide and 0.5 m deep. **Left**: A wadi west of Soada/Suweida filled with water after rainfall. ©Hayan Hmidan, cc-by-2.0 [79]. **Right**: Canal of Kharsa/Salakhed (east of Bostra). [59] (Figure 16).

I show these relationships with the example of Seeia and Kanatha (Figures 2 and 4, labeled as Qanawat and Si', respectively) [74] (pp. 393–396)]. According to the remains and sources, the two apparently have connections to other places via water distribution. This demonstrates how these relationships created a system of partners rather than hierarchies: Seeia, as a sanctuary and settlement on a spur in the Jebel (1300 m a.s.l.), from the first century BCE, received water from the Jebel and stored it in cisterns [78] (esp. Figure 4) [80–82]. Also, a water distribution facility was located in front of the sanctuary. Many underground channels can be traced between Seeia and Kanatha, two kilometers to the north, leading to reservoirs and cisterns in Kanatha [83]. Kanatha had features of a city—a city wall, various temple buildings, a theater and a nymphaeum—that corresponded with its 'upgrading' to the status of a *polis* in the second century CE. In particular, one of its sanctuaries was prepared for storing huge amounts of water (Figure 6) [78] (see Figure 4). The temple of Zeus Megistos overlooked the large sacred area at the highest point of the city [82] (see reconstruction Taf. 85, Beil. 8).

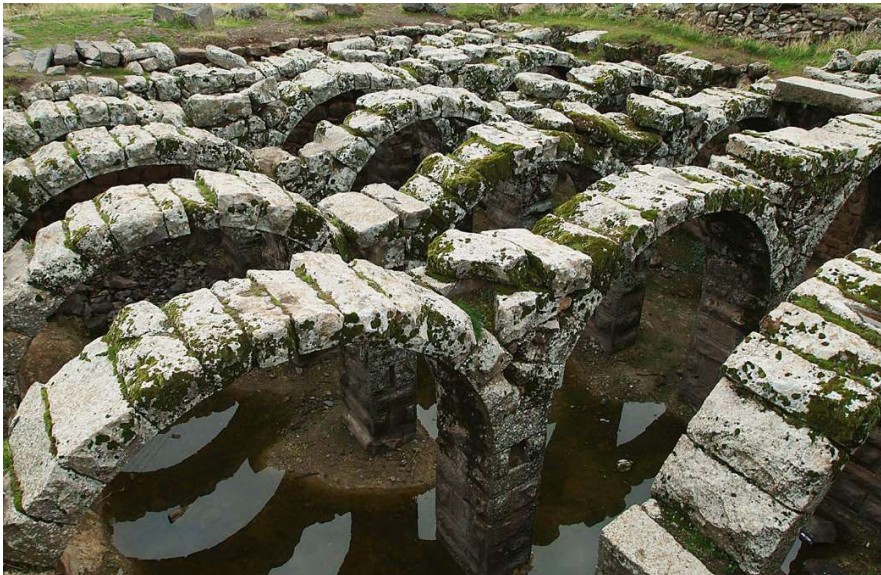

**Figure 6.** Cistern in front of the temple of Zeus Megistos at Kanatha. ©James Gordon, cc-by-2.0 [84].

Seeia is normally treated as an important sanctuary and meeting place, and due to its vicinity to Kanatha, is considered to be dependent on that city. In terms of the water management system, the situation is reversed (or at least more balanced): Kanatha has no water without Seeia. Yet, the dependencies also extend to other spatial scales. At Kerak, some 20 km west of Kanatha, the inhabitants worshipped the god Zeus. However, this was not the Zeus of their village and community, but the *Zeus Megistos Kanathetôn*, precisely of the people of Kanatha (Figure 7) [85] (no. 9810. 9799). The inscriptions can be roughly dated to the Roman period (second or third century CE); however, the temple of Zeus at Kanatha from the third century CE presumably had a predecessor dating to the first century BC/first century CE ([54], ch. 3.4.7 and 3.4.8) Here, the dependency appears even stronger than that between the settlement and the communities of Seeia and Kanatha (Figure 8).

**Figure 7.** Inscriptions pointing to bi-directional relationships between the settlements. (**a**) The people from Kerak in the plain of Nuqra worship Zeus Megistos at Kanatha, located 20 km to the east of Kanatha, from where it received water. [86] (no. 2412d), [85] (no. 9799). (**b**) Inscription from Aphetata/el-Afineh about the channels of water (*agogoi hydatôn*) from Kanatha. [86] (no. 2296. 2297).

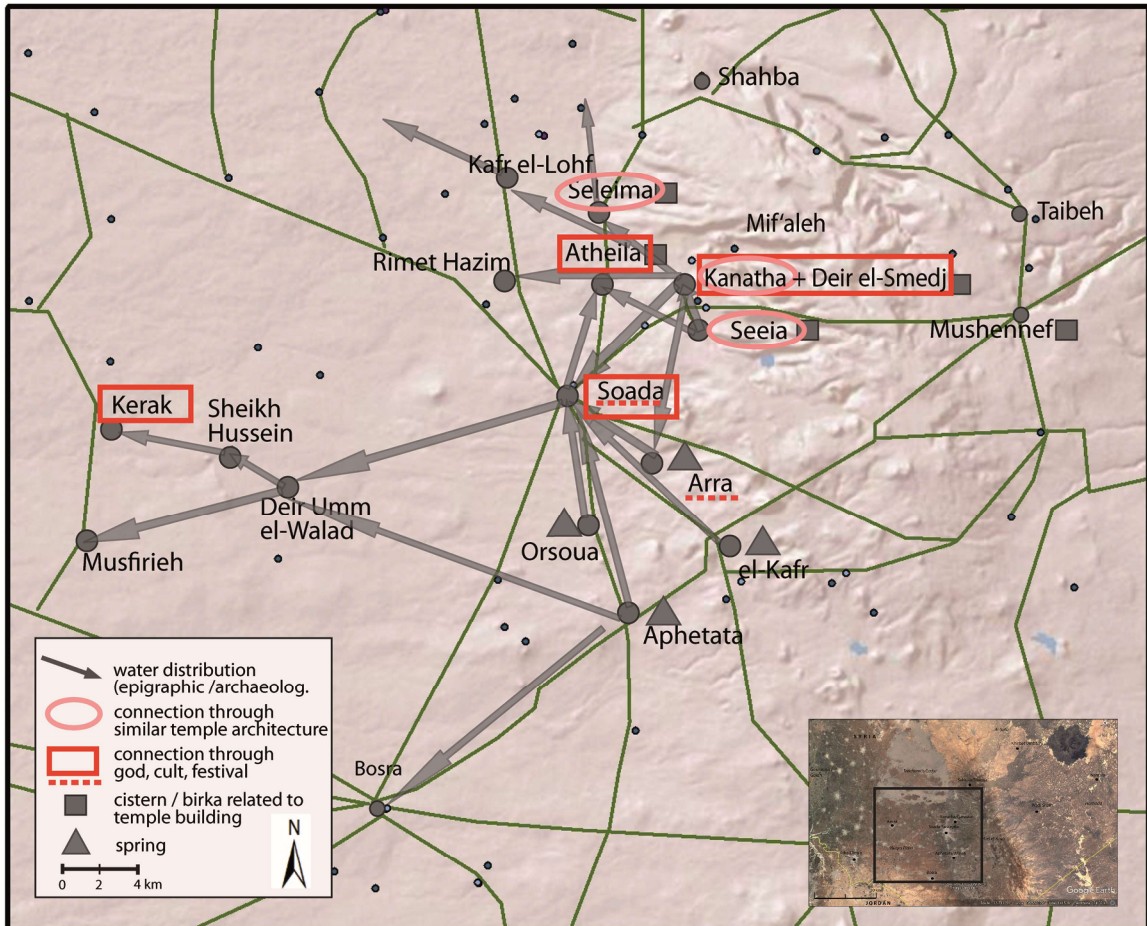

**Figure 8.** The relationships between landscape, resource, settlements, and people created through water management (mono-directional) and religious institutions (bi-directional) in the western slopes of Jebel el-Arab and the plain of Nuqrah. Map: A.-K. Rieger.

The settlement of Soada to the west of Jebel el-Arab is an example of a central place in the region due to various functions, with city status from the mid-second century CE onward, and its urban features (nymphaeum, colonnaded roads, and theater) [87]. Some villages to the west received water from Soada (and through Soada from Kanatha): Museifireh, Deir Umm Walad, Sheik Hussein, and Kerak [86] (no. 9810. 9811. 9815. 9817). However, Soada itself depends heavily on the water resources from Jebel. It is a recipient, which is a weak position. First of all, Kanatha, but also other places to the south—Arra, Aphetata, and Orsua—send water to Soada [85] (no. 2296. 2297. 2308) (Figure 9). To be on good terms with those settlements and communities that were important to Soada, the citizens or city council of Soada sponsored a temple and a statue of Athena at Arra [85] (no. 2308) (Figure 9). The relationship of the people of Soada to those of Kanatha was re-instantiated every year in a common festival. At Deir el-Smedj on the outskirts of Kanatha lies a huge temenos where an inscription was found, telling us that the people of Soada financed a communal festival [85] (no. 2374a; [88] (no. 144. 171). The cases from Kanatha, Soada, Kerak, and Arra show that the water distribution system and the guarantee of water availability functioned as a reciprocal system. Deities as transcendent protectors (Zeus and Athena) were called upon to supervise the organization and management. At the settlement of Aphetata, one of those directing water to Soada, various lines of water supply are connected. The source at Kôm Nebe Ira (Figure 4) was most likely tapped for water for the area of Soada. Other sources in the area of Aphetata (e.g., Nimreh Qraye) presumably fed the aqueduct running southwest to Bostra [60] (p. 103, 134–135) and [78]. Based on the elevation of Aphetata,

at 1100 m a.s.l., is one of the lowest points sending water to Soada, and the watersheds between the individual wadi catchments are the interface between the northern and southern catchments (Figure 8).

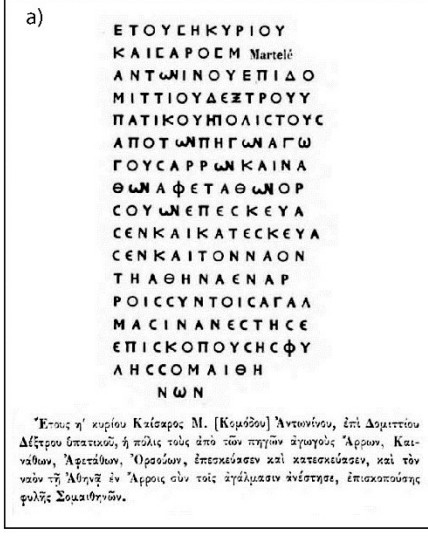

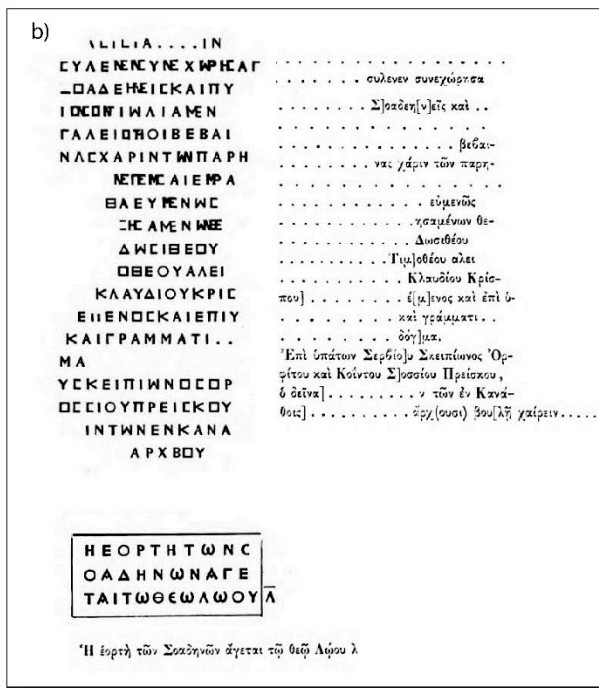

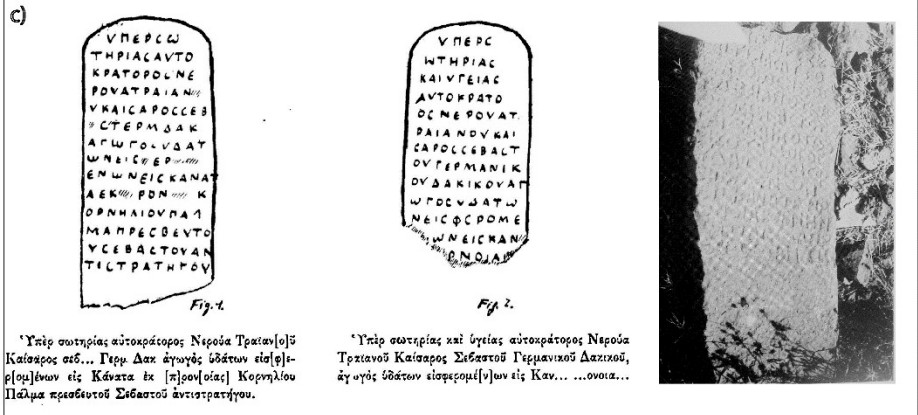

**Figure 9.** Inscriptions pointing to bi-directional relationships between the settlements. (**a**) Inscription from Soada about the renovation of water channels (*agogoi*) from Arra, Aphetata, and Orsua, and the construction of a temple and a statue of Athena by the *demos* of Soada at Arra. [85] (no. 2308). (**b**) Inscriptions on stelai from Soada (left) and el-Arra (center) and the stela from Zawiyet Balahat, speaking about the water that was directed from Kanatha to Soada. [89] (Figure 1. 2). (**c**) Inscriptions from Soada (left) and Dmeir el-Smedj (right) about a feast at Dmeir el-Smedj involving both communities. [85] (no. 2307. 2370).

For the provincial capital and the city of Bostra, we do not have evidence of relationships with other places that were established and confirmed by religious institutions. However, the water on which Bostra relied, with its baths, nymphaea, and a considerable population, originated from the Jebel to the northeast [87,90,91]. To determine the centrality depends on the degree of independence a place has. The capital is not independent from this huge area on the slopes of Jebel el-Arab, with its sources, settlements, and communities, where traces of a dominant administrative control cannot be found (Figure 8). Moreover, the Roman provincial border between Syria and Arabia runs across these

watersheds and drainage systems that supplied Bostra. Juridical implications and dependencies came certainly along with this territorial particularity.

Resulting from the perspective on relationships through water management and religion is the concept of central places and spaces they controlled is not applicable to Hauran. Even though the settlements are nuclei of housing, religious institutions, administrative, and economic infrastructure (reservoirs, roads, markets, meeting places, even baths, etc.), they cannot be considered as independently acting. The imagery of a rhizomatic system, in the sense of Deleuze and Guattari, is more suitable for understanding the place-space-resource relationships at work in Graeco-Roman Hauran [92]. The French philosophers used the image of a rhizomatic, web-like root system from botany for describing and organizing the phenomenology of the world in contrast to the (structuralist) tree-like organization. The various individual settlements appear on the surface, but exist only due to their rhizomatic entanglement with the other settlements and sources. The relationship is not mono-directional, since water is directed down the slopes, but people were connected up the slopes by religious practices. To concentrate on one central place is not warranted. All places partaking in the described network were more or less central according to the various parameters of vicinity to water, vicinity to encampments, exchange with nomadically living groups, closeness to routes, and the number of inhabitants and facilities. However, there was no center of this kind that could have existed or survived without the close and re-established contact to the other settlements.

## 5.2. Marmarica

Marmarica exemplifies, from a different viewpoint, the importance and value of marginal and un-central areas. This arid region, between the Nile Valley and Cyrenaica on the fringes of the Libyan Desert, was underrepresented in research up to the late 20th century CE. Only Siwa in the south and also the northern parts have drawn the attention of travelers, archaeologists, and anthropologists (Figure 3) [93–95]. However, it is not only due to the little interest in the region but also its character that only few find materials and remains are known. People living here from the Bronze Age to Graeco-Roman times followed combined strategies of livestock breeding and agriculture, and were therefore prepared for both drought years and good years. However, landscape archaeological studies, pursuing a spatially large scale and diachronic approach, broadened perspectives to socio-spatial organization, economic surplus production, as well as to intra- and inter-regional connections that helped with understanding how the region functioned in antiquity in a way that was far from 'marginal' (Figure 10) [61,63,65,66].

The first issue for human, faunal, or floral life in Marmarica was the scarcity of water and the poor quantity and quality of soils. Apart from the 150 mm annual rainfall in the coastal strip, the entire region has to cope with arid conditions. Rainfall decreases with distance to the coast (Figures 11 and 12). Only with water and soil harvesting measures on the tableland, along and in the wadis incised in the tableland, can the conditions for agriculture be generated (Figure 12). People from the second millennium BCE onward built and amassed embankments, terraces, and dams with the field stones at hand. Soil was accumulated behind terrace walls, dams, or shallow embankments by the flow of water by which it was transported (Figure 13). A peak of managing and harvesting water and agricultural production occurred in the first to the fourth century CE, in Roman times.

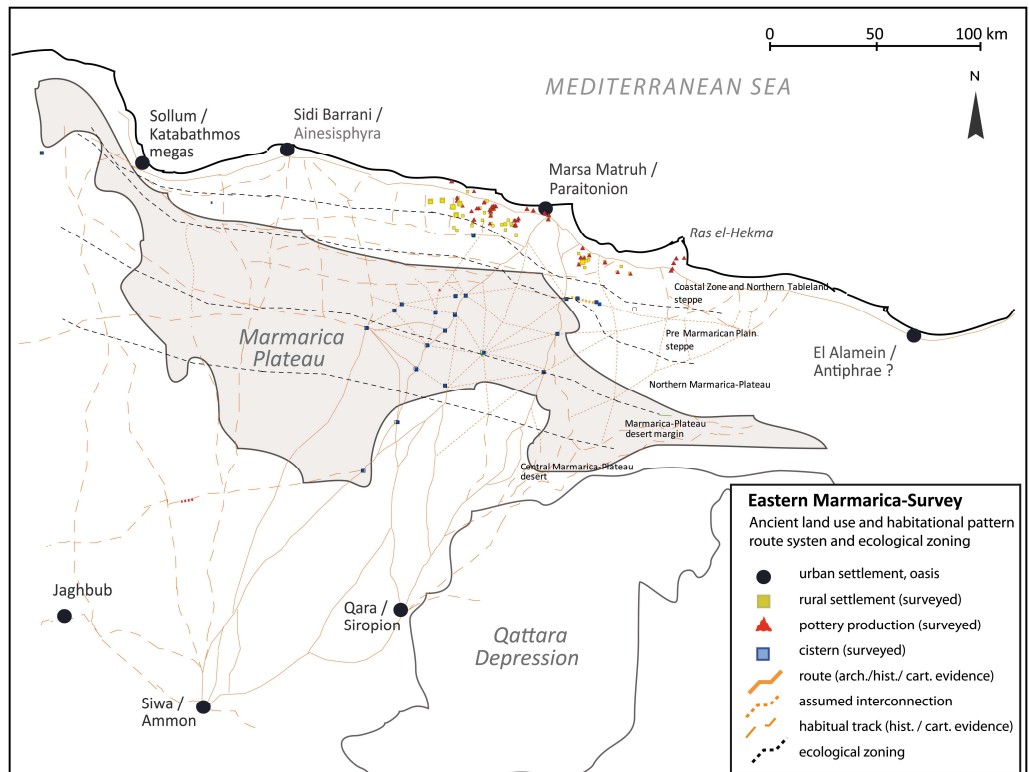

**Figure 10.** Map of the eastern parts of Marmarica showing habitational and land use patterns, production areas, and the route system in the ecological zoning. (Minor findspots like campsites or fields cannot be displayed on this scale.). Map: A.-K. Rieger, T. Vetter.

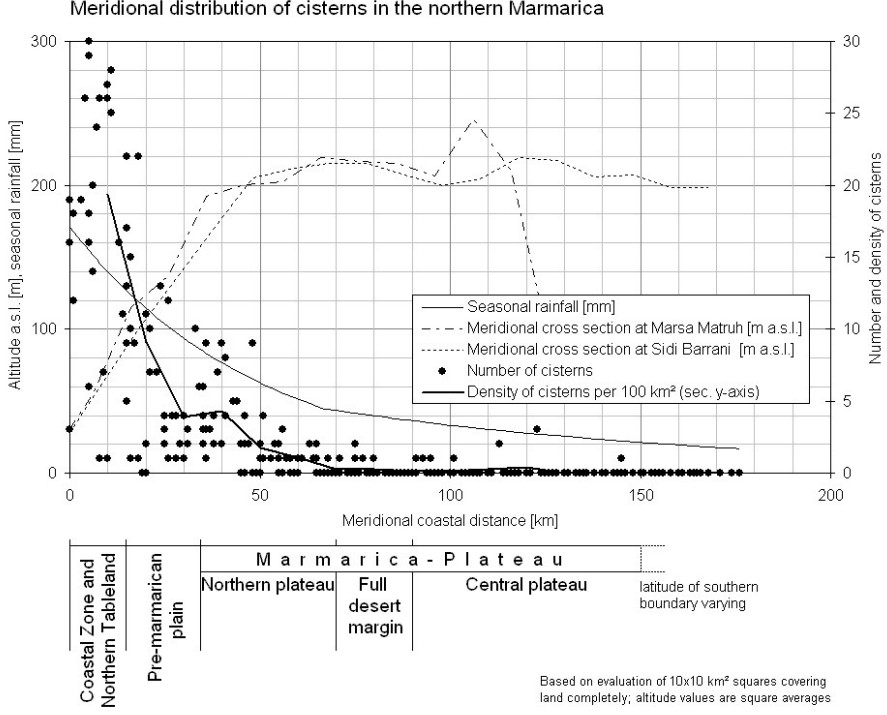

**Figure 11.** Cistern density in the area between Sidi Barrani/Ainesisphyra and Marsa Matruh/Paraitonion in recent times in correlation to mean rainfall, relief, and coastal distance. A high number of the cisterns on the tableland date back to Roman times, whereas the surveyed cistern sites on the Marmarica Plateau show evidence of older periods. Graph: T. Vetter.

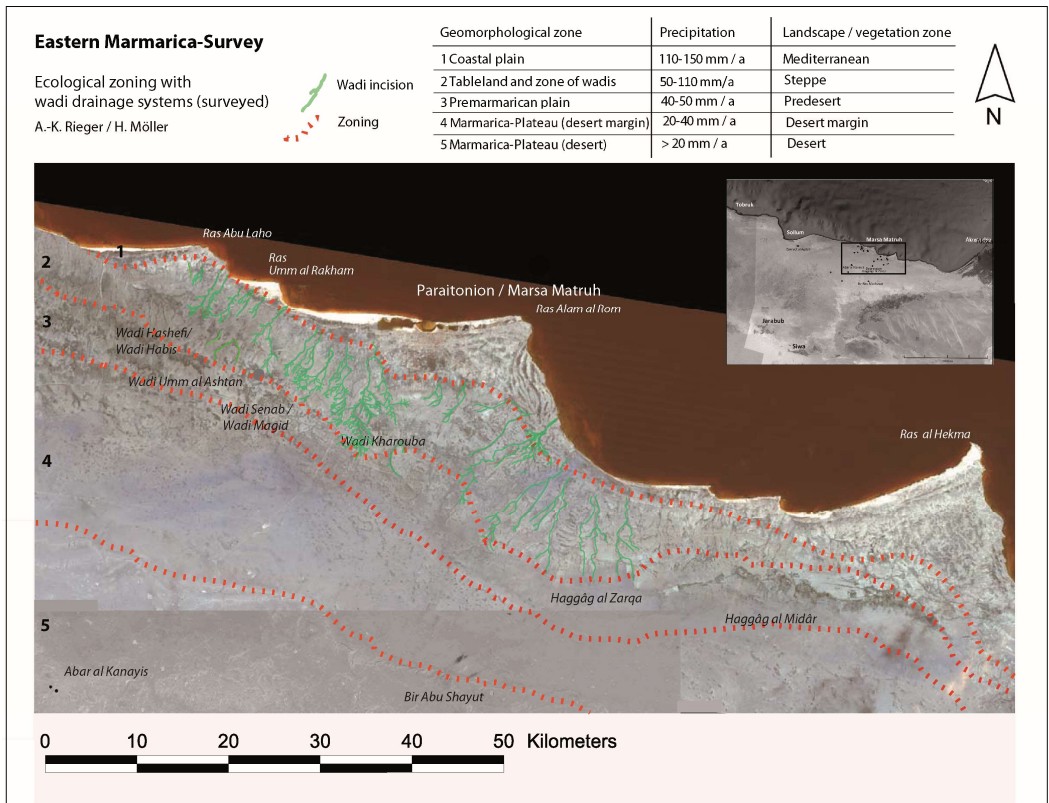

**Figure 12.** Map of Eastern Marmarica showing the ecological zoning, the wadi drainage systems, and the mean rainfall. Map: H. Möller, A.-K. Rieger based on Landsat 5TM 179–38.

Water from overland runoff from mainly winter precipitation can be stored in the soil. This "harvested" soil and water allows for the growth of crops and trees (Figure 14) [62,63]. All agricultural production is determined by the amount, direction, and velocity of the water. By these means, the cultivable land amounted to no more than 9% of the tableland in Graeco-Roman times (Figure 3). However, fallows or the crops on the fields in drought years could be used as grazing areas, as was the case with the steppe zone south of the tableland. The yields and returns from agricultural production (barley, grapes, figs, and only little wheat) and livestock breeding amounted to a surplus, whereas marginal areas are normally considered to only allow the inhabitants a subsistence economy (Figure 15) [62–65,96]. This surplus production was reconstructed mainly from the existence of numerous pottery production sites along the coast and on the tableland dated to between the second century BCE to the fifth century CE, and from the peak in the number of settlements (Figure 16) [64,65] (p. 144). The production of the potters' workshops was mainly transport amphorae [97]. These locally-produced amphorae were at least transported to the south, to Siwa, as demonstrated by the findings at water supply points and the route network on the Marmarica Plateau [63,66].

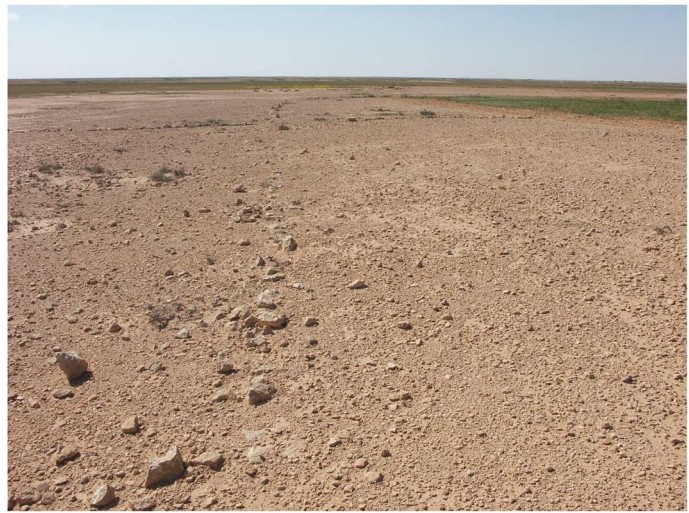

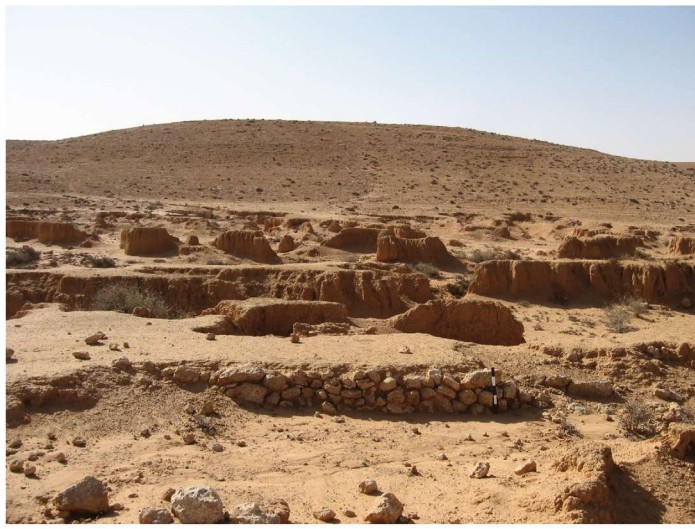

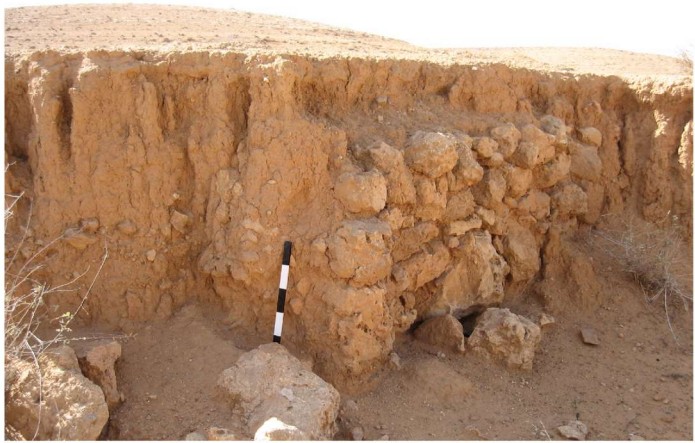

**Figure 13.** Embankment on the tableland for creating cultivable areas (Hâggag Midâr) (**top**); cross sectional dams in the bed of Wadi Kharouba, section (**middle**); surface view on a dam/terrace in the bed of Wadi Kharouba (**below**). Photographs: A.-K. Rieger.

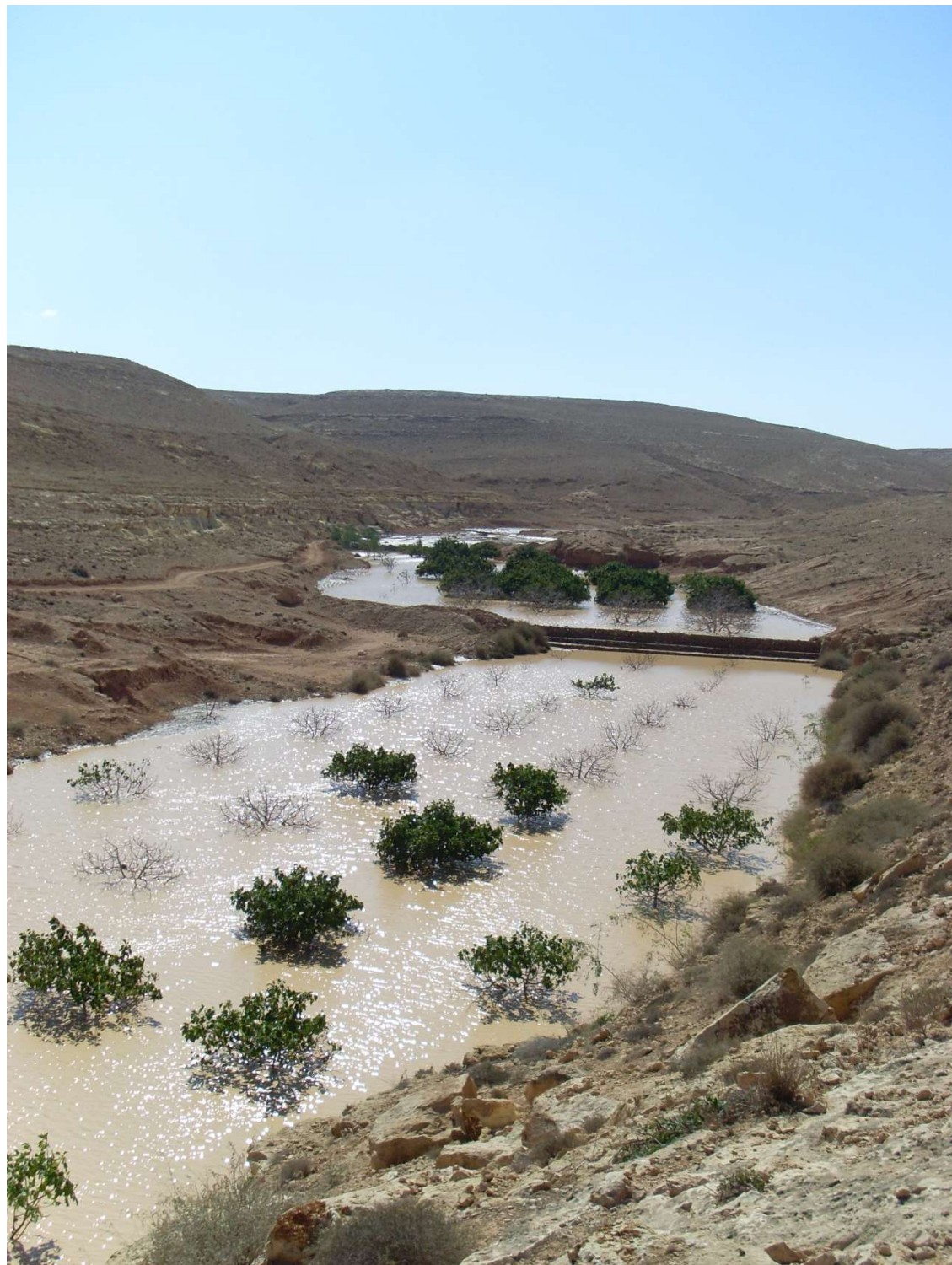

**Figure 14.** Wadi bed with fig trees and ponding water behind the terraces after rainfall. Photograph: A. Nicolay.

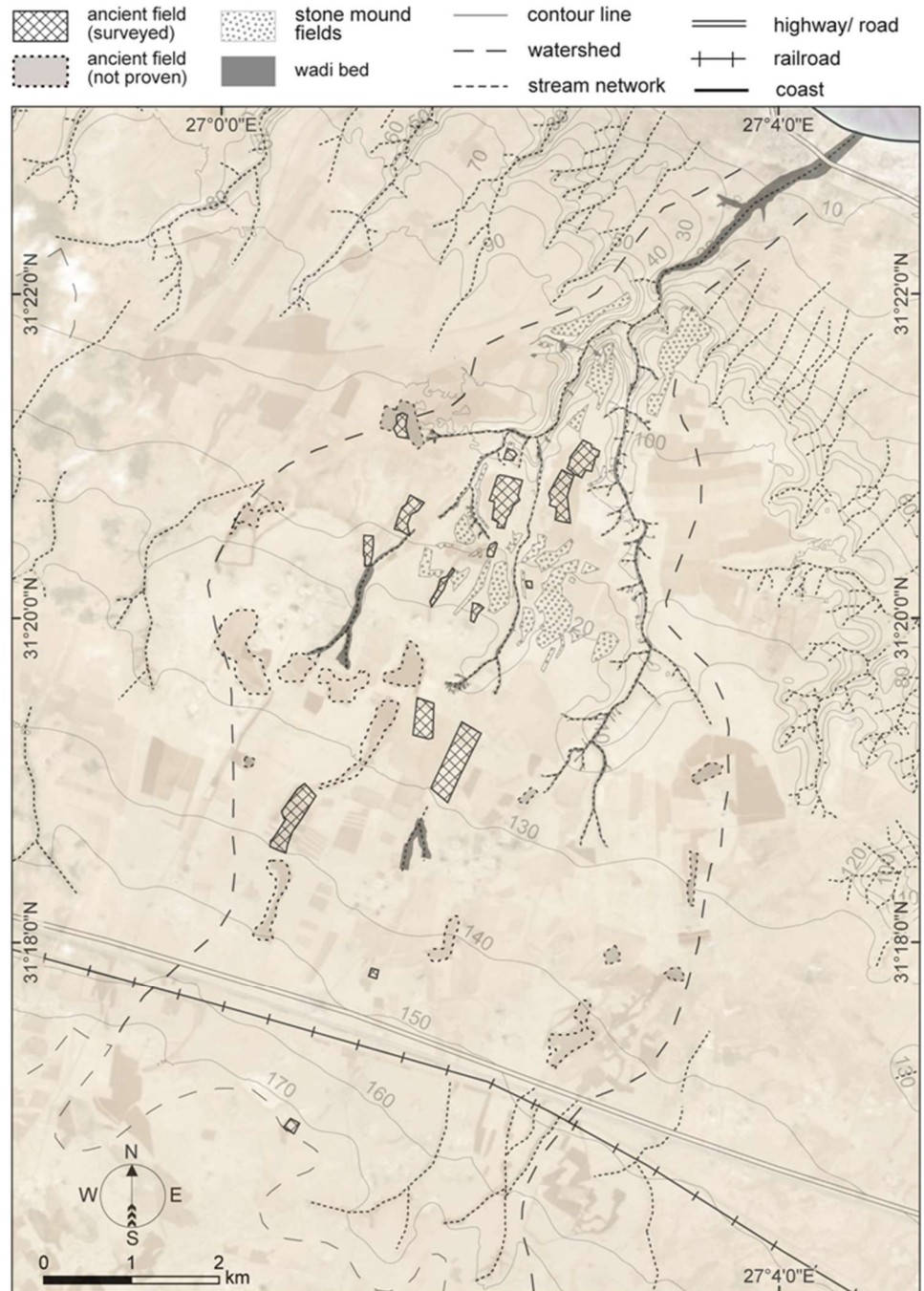

**Figure 15.** Watershed/drainage system of one wadi (Umm el-Ashdan, cf. Figure 12 for its location west of Marsa Matruh) with the various cultivable areas. Map: A. Nicolay, A.-K. Rieger, T. Vetter.

Resource availability preconditions the economic life strategies and the habitational pattern, which was also true for Graeco-Roman Marmarica. The habitational pattern is structured according to water availability: In the south (Marmarica Plateau), only water supply points (cisterns) and campsites close by or in depressions with some vegetation for grazing occur (e.g., Abar el-Kanayis, Abar Abu Mukhayat) (Figures 10 and 12) [63,65,66,98]. Campsites, fire places, and other human traces between cisterns and depressions have only been rarely traced. The steppe zone (Premarmarican Plain) allowed agricultural use in limited favorable locations, but was sustainable only when combined with livestock breeding. In this zone, campsites, such as those of herders, have also been traced. This trend continues to the north in the zone of the tableland and the wadis where the highest density of settlements

occurred, with different amounts between two to three farmsteads up to 25 to 30 farmsteads with pottery production sites of various sizes [64].

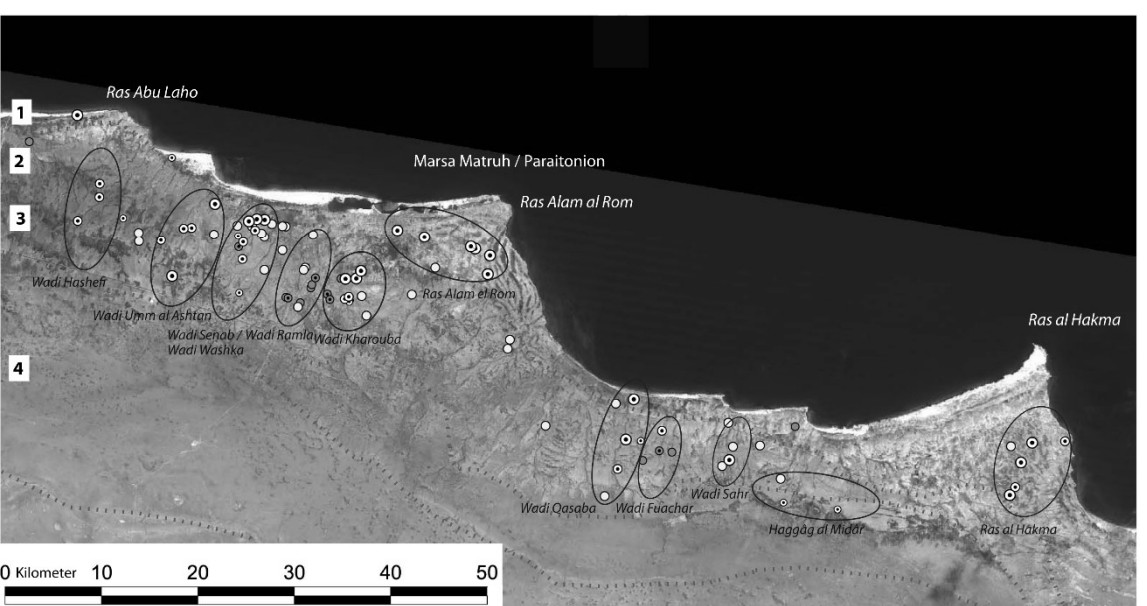

**Figure 16.** Pottery production sites in Eastern Marmarica. Map: H. Möller; A.-K. Rieger based on Landsat 5TM 179-38.

The course of the water structured the entire region, which had harbors and anchorage points along the coast, and with the city of Paraitonion (modern Marsa Matruh), created an administrative and economic center (Figure 12). However, the coastal strip was only the last link in the flow of the water. Due to the slight but sufficient inclination of the Marmarica Plateau in the south, the runoff reached the catchment areas of the wadis on the tableland. The hydrological regime depended on the water coming from the south [99]. The parallel drainage systems of the wadis themselves, and the settlements along the escarpment, which divides the coastal strip from the tableland, do not depend on each other, even if they are spatially much closer than the plateau in the south. All activities and the associated settlements, water supply points, grazing areas, and production sites depend on the runoff-conditions on the plateau. So, the orientation of those living on the tableland was oriented toward the south, as was the case for those living in the coastal zone. If people on the plateau, the tableland, and along the coast did not join forces and invest labor into a correct water management system in the southern parts, neither could water be harvested nor soil be accumulated to store the water between, along, and in the wadis or in the coastal plain.

Since water management was necessary not only to obtain water, but also to limit it, flood prevention was an issue. Investment in water management was as important for those upstream partners as the water itself was for the downstream partners in the system. Due to the relief of the region, this south–north structuring continued across to the plain along the coast. The people living in the coastal plain—as the final and lowest members in this chain of water—had a strong interest in suitable water management in order to prevent fertile soils from being washed away, but simultaneously maintained high and stable water availability.

Accordingly, the organization of the land and the people living on it was structured by wadis in line with the south–north running water. This means rather independent sub-systems in the tableland and wadis zone ran south–north (Figure 10). The water management only functioned if all

agents in one catchment area along the line of the runoff and along the wadis participated equally. This causal correlation of the upstream and downstream neighbors can be considered evidence for the non-centralized organization of socio-economic life in Eastern Marmarica.

Even if the control of resources promised power—the more one controls, the more one's power increases—the management of scarce resources led the people in Eastern Marmarica to collaborative rather than centralizing measures. The people along the coast were dependent on the upland management of the water. Comparable collaborative investment was true for oases in the Libyan Desert (e.g., Kharga), which led to a heterarchical system of socio-economic organization, whereas the concentration of power often results in hierarchical systems.

In the case of Eastern Marmarica, one could think of the elite of landowners as known from the Roman landscapes of the Nile Valley or the western North African regions. By analogy, this conclusion does not consider the local conditions. As laid out above, it was rather the collaborative investments and heterarchical organization that allowed people to live in Marmarica in Graeco-Roman times.

The approach of reading the landscape of the Marmarica in an un-central way does not mean that the existence of central places should be denied. There were central places like the Oasis of Siwa and the harbor of Paraitonion, which were places where economic and political powers were (and are) concentrated. Yet, the factor of time should be considered. Temporary central places can be markets according to harvesting periods, pottery production sites according to the demand of storing or trading agricultural goods, the rhythms of caravans, and the need for certain goods (meat or grain) on a regular basis, all of which offer a more dynamic and organic picture of when and how the oases or the harbor cities had a central position and significance.

## 6. Conclusions

The choice of a marginal landscape characterized by aridity requires a more nuanced and differentiated application of central place theory based on different environmental and cultural settings. The combination of models, such as the small worlds phenomenon and weak and strong ties from network analysis with what socio-spatial practices, such as the land use and habitations that we find in these two ancient arid landscapes, broadens the perspective of arid landscapes and lifestyles of people. Researching the issue of rural water management as a non-static resource has proven to be useful for overcoming overly site-oriented approaches of past societies, which mirrors the often underlying influence of hierarchical thinking as propagated by (traditional) central place theory. Dynamic views, focusing on the interactions and qualities of relationships, which, in the case studies, included the methods of the resource management and involved people, places, and spaces, provide new perspectives beyond the model of central places regarding life strategies, land use patterns, and social organization. Arguing in favor of the non-centrality of landscapes ("un-central landscapes"), or of landscapes as places, we are able to blur the line between the center and the surroundings. Landscape archaeology—among its multi-layered merits—focuses on larger areas and not only on sites. The benefit is that archaeologists can enhance their ability to consider contexts, interrelations, and their complexity. Many conceptualizations of marginality have been derived from the study interest in cities—sites of condensed material evidence, and facilities—wherever and whenever they were located. The marginality approach combined with a more differentiated understanding of central place theory applies more clearly to areas and to networks, and does not define cities as the *non plus ultra* of socio-cultural and economical human activity.

Resulting from the case studies of arid, ecologically marginal landscapes of Graeco-Roman antiquity are the following observations:

1.  Un-central $\neq$ unimportant $\neq$ no potential The case studies presented non-central areas when seen from the perspective of the Graeco-Roman Mediterranean. They were/are marginal in the sense that they were characterized by ecological marginality (arid environment) and economic marginality, which is associated with social and political marginality. However, the studies

demonstrate a strong position in a network of relationships, and also an economic potential that was considerable in relation to ecological conditions, and the yielded surpluses.

2.  Un-central = not site-oriented, but area-oriented Methodologically, thinking in un-central terms helps viewing historical phases, cultural phenomena, or economic relationships using a non-site-oriented method. Landscape archaeological, area-oriented approaches should be much more relevant in research design to go beyond the fixation on sites. In the case of Hauran, the area- and relationships-oriented approach, as presented here, can better explain the settlement pattern of the region, where research 'sticks' to villages and their temple buildings. In Marmarica, only an area-oriented view provides results due to the large areas with which habitational areas were connected in order to receive water.

3.  Un-central = marginal = sensitive and resilient = historically of interest Especially in ecologically (and economically) marginal areas, the sensitivity and can be more pronounced and recognizable than in ecologically well-suited regions. In regions at the margins, inhabitants were used to a fragile balance, to good and bad years, to necessarily adapting to scarce or overly abundant resources. Hence, in the case of Hauran and Marmarica, climatic changes or crises, political developments, and modifications in economic relationships could either have strong and fast effects due to the fragile situation, or could not affect areas and people according to their abilities to more easily and quickly adapt to changing situations. For this reason, the evidence from (arid) marginal regions can reflect larger (global in the sense of the MENA-region) historical developments, which are not easily visible in areas of complex, dense, and politically biased sources. The socio-economic (and socio-religious) history of marginal regions can archaeologically appear in a better resolution of contexts, findings, remains, and soil.

4.  Un-central ≠ no complex interdependencies (people, resources, spaces) An un-central approach to the study of past societies and economical systems is suitable for gaining insights into the complex interdependencies of people, resources, spaces, and political or natural changes. Investigating marginal areas of Marmarica and Hauran offers insights into the complexity and the connectivity, provided that we accept different ways of living, such as mobile life-strategies, the particularities of communities and individuals involved in trade interactions. These are methodologically difficult to grasp but are not marginal players in marginal regions.

5.  Un-central areas = marginal, but from two sides = areas 'in-between' Un-central areas, as presented in the two case studies, are zones in-between politically and economically often more powerful areas. Their role as buffers, areas of contact (trade) but also of conflict (control, war), is of significance for political and economic balance. Both Hauran and Marmarica—though not the main issue in this contribution—played a mediating role in Graeco-Roman history and beyond. Yet, acknowledging the other side of the margin is as enlightening as the area-oriented approach outlined above.

6.  Un-central = temporal shifts in degrees of centrality Places can change their centrality in the course of, for example, an agricultural period. Temporal shifts and oscillations should be considered when determining the centrality of places. This methodologically challenging point adds more fluidity to a rather stable concept. The cases of Hauran and Marmarica started from water, which is a fluid and not a regularly available resource. As such, the approach considers the changing status of places and the multiple roles. A place receiving water is also a redistributor (Soada). Political central places like Bostra were highly dependent on a huge area. Which places were more central has to be answered in a differentiated way by investigating their web of relationships. A temporarily changing use and varying degrees of places being frequented, according to economic flows of harvest, or periods of processing goods, and best times for trading goods, can be postulated in Marmarica. Cisterns along routes, potters' workshops, and harbor sites were not continuously central on the respective scale.

7.  Un-central areas ≠ not deprived of centers The definition of a what is perceived as the center should be clearly explained in every single study, depending on the material categories and

the scales being examined, and should be integrated into an interpretation. The centers in both Marmarica and Hauran are those with a high concentration of functionalities and features, of people and power—even if they were not perceived as such all year round. Yet, their position in the overall socio-economic or socio-religious organization varied.

From the different quality or intensity of relationships represented by material remains in the two ecologically marginal regions, we can reconstruct the socio-economic and spatial frames of interactions, the connectivities, as well as the catchment areas, and integrate these elements into a perception and construction of these arid landscapes through these interrelations [16] (pp. 44–46).

The case studies of Graeco-Roman marginal regions, Hauran and Eastern Marmarica, showed that their water management and socio-economic organization were not centralized issues. Exploring the interactions resulting from water management forces an analysis of the spatial scales, the qualities of interactions in the geo-physical, as well as social landscapes.

People living there had to deal with it in ways and modes involving close social interaction, which, at the same time, spanned large distances and spaces. Strong ties as well as weak ties were at work between the people, resources, gods, and places. In Hauran, we found deities, festivals, and religious offices together with physically built and maintained water channels as eloquent evidence of the relationships of assumedly independent settlements, without one being more central than the other.

In Eastern Marmarica, the direction and varying quantities of runoff water together with morphological characteristics led to a collaborative rather than hierarchical organization of water harvesting and distribution and shares of cultivable land. It is not a nucleus with a surrounding environment that can be used to visualize the socio-spatial organization reconstructed by landscape archaeological methods, but rather a rhizomatic (Hauran), and a dendritic (Marmarica) structure.

**Funding:** The data and approaches that were integrated into a comparative interpretation were sourced from parts of two different research projects. The first is the "Marmarica Survey" funded by the German Research Foundation (2004-08 as part of the CRC 586 "Difference and Integration"; 2008-11 as independent project of the author) at the University of Halle-Wittenberg. The second project, of which Hauran was a part, was funded as the sub-project "Enlivened spaces. Spatial patterns and social interactions in sacred contexts of the Roman Near East" of the ERC Advanced Grant in the FP7 (Agreement No 295555) "Lived Ancient Religion", directed by J. Rüpke and R. Raja located at the Max Weber Centre for Advanced Cultural and Social Studies at the University of Erfurt, Germany.

**Acknowledgments:** The editors deserve credit and thanks for all their efforts and support for writing and publishing this article in their Special Issue. The reviewers' comments helped significantly to enhance the argument. Without the intellectual support of Gunnar Brands, Valentino Gasparini, Heike Möller, Alexander Nicolay, Georgia Petridou, Rubina Raja, Jörg Rüpke, Thomas Vetter, and Jutta Vinzent, no results could ever have been obtained. The Supreme Council of Antiquities in Egypt supported our work in the Marmarica constantly. Adam Newton corrected the English.

**Conflicts of Interest:** The author declares no conflict of interest.

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
