# Peer review of "‘Un-Central’ Landscapes of NE-Africa and W-Asia—Landscape Archaeology as a Tool for Socio-Economic History in Arid Landscapes"

_land, doi:10.3390/land8010001_

Reviewer 1 Report

Review of „‘Un-central‘ landscapes of NE-Africa and W-Asia – landscape archaeology as tool for socio-economic history in arid landscapes submitted to the journal Land 2018, 7

General remarks

This paper presents two highly interesting landscape archaeological case studies that were studied independently in different contexts by the author. The first case study is geographically limited to the Hauran desert in southern Syria and northern Jordan dealing with water management issues between urban “centers” (e.g. Kanatha) as well as major religious sites (e.g. the temple at Si) (pp. 9-14). The second case study is set in the Mamarica (NW-Egypt) and researches rural (desert) water management strategies that are not connected to any major sites (pp. 14-21). While the two case studies are situated in two different geographical (and to some extent also cultural) areas, there is common ground that makes a comparative study (as presented in this paper) fruitful. First, both studies roughly deal with the same time period (Graeco-Roman). More importantly, however, both are set in an arid, desert environment and the archaeological evidence discussed in both cases (structures and features related to water management) derives from a predominantly rural environment.

The author is therefore dealing with so-called “marginal” or “un-central” landscapes (from both an environmental as well as infrastructural perspective). For this reason, it is certainly correct (as the author has nicely achieved) to, first, briefly lay forward a critical and more differentiated discussion of Central Place Theory (referred to as CPT) as well as the concept of marginality in archaeological and historical research. Interestingly, the author criticizes the more traditional understanding of CPT as being too site-oriented and reveals an inherent neglect of areas (or landscapes) within this theoretical frame (pp. 1-2). The author thus calls for a more “un-central” way of thinking (pp. 2-3) and to include the “areas between” central places in farther-reaching archaeological and historical studies in order to comprehend the “central places” and so-called “un-central” landscapes comprehensively in their respective environmental and culture-historical context.

Together with a strong landscape archaeological methodology and a revised definition of the concept of marginality (pp. 6-8) as well approaches derived essentially from network theory (pp. 8-9), the author convincingly concludes (pp. 21-22) that assumed “un-central” areas are indeed important and offer many potentials for contextualizing and better understanding “places” in arid, “marginal” environments. For example, the Hauran example clearly shows that urban centers (Kanatha) and religious sites (Si) were immensely dependent on the successful organization of their rural environs – particularly the overall water management – and vice versa. The dependencies were bi-(or even multi-)directional. Additionally, the author’s studies on the rural water management in the Marmarica has demonstrated a clear system of co-dependencies that were (and still are) completely divorced from any central place or authority and was instead developed to secure more sustainable subsistence strategies (a combined pastoral and agricultural lifestyle) in such an arid, desert environment.

This paper is thus both timely and original – not only regarding the important insights gained into the rural water management systems of the Hauran and the Marmarica, but touches important theoretical issues within current (landscape) archaeological discussions: It convincingly argues for and nicely demonstrates the potential and information that may be derived when not only considering central places, but also perceived “un-central” landscapes.

I therefore highly recommend this paper for publication within the journal Land.

However, there are some important remarks and comments that I would greatly suggest to follow before publication. This criticism is to be understood entirely constructively in the hope of enhancing the overall quality of the paper. The following shall summarize the main issues, but I suggest to carefully consider the comments and markings in the text (attached). In the text, red markings highlight where I noticed lingual errors, spelling or syntactical issues or otherwise unclear or misleading passages. Yellow markings signal comments related to the content or structure of the relevant passage.

Thank you for this interesting paper!

Structure

The paper is generally well structured and can be followed. However, I do have some remarks which I would suggest to seriously consider as this may significantly help the overall understanding of the paper and facilitate the flow for the reader (compare also the comments in the text).

- Section 1 and 2 are clear as they nicely introduce the reader into the (critical) theoretical background of CPT and the “un-central way of thinking”. However, the transition to the definition of aridity and marginality (p. 3) comes somewhat abrupt. I would therefore suggest to more elegantly “smooth” into this part.

- Most importantly, however, I do not fully understand why section 3.1 “Two arid regions – but are they marginal regions?” is placed between section 2 (“Thinking un-centrally”) and section 3.2 “Marginality – a concept to be differentiated”) as well as section 3.3 “Weak and strong ties and the study of marginal areas”.

Sections 1, 2, 3.2 and 3.3. are more methodological and theoretical in nature while section 3.1. seems to introduce more into the (marginal) arid environmental conditions and the (brief) historical context of the two case studies.

In order to facilitate the readers understanding of the text, I would therefore strongly suggest to set section 3.1. as an introduction to section 4 (where the two case studies are presented in more detail). This way there would be a clear “methodological-theoretical block” at the beginning of the paper (sections 1, 2, 3.2 and 3.3) and a clear section where the two case studies are laid forward (section 4 – with 3.1 as a “introduction”).

- Another structural comment concerns the conclusions (which are generally agreeable). The listing of the observations made on the basis of the case studies is very useful. However, I do feel that the argumentation will become clearer to the reader when EITHER some examples from the case studies will be added to each point listed in the conclusion (points 1-7). AND/ OR the passage following the points (which concludes the main points discussed for the two case studies) could be moved before the observations. This way the reader is led from more detailed conclusions related to the case studies to more general remarks concerning the (landscape) archaeological research of so-called marginal or “un-central” landscapes.

Language and Style

Only few spelling mistakes were noticed. Unfortunately, however, some major syntactical and formulation irregularities/ errors were noticed, which greatly impeded a smooth read and the overall understanding of the text.

It is strongly advised that the entire text is (again) carefully and comprehensively proof-read by an English native. I have highlighted all passages in the text in red, which are in need of essential re-phrasing. However, this is not entirely systematic and shall not replace thorough English language checking.

Importantly, please avoid long and complicated sentences – this only impedes the reading unnecessarily; especially when combined with many sub-clauses etc. Keep sentences short and to-the-point.

Finally, parts of the conclusions appear to have been written in haste or seem unfinished. For example:

- p. 21 (line 602): “[…] economic potentials that are on their scale XXX, considerable, and offered surpluses.”

or

- p. 21 (line 608): “”Umland Surveys” = still starting from the center”

Figures

The majority of the figures can be read well and illustrate the paper nicely. However, there are a few figures that require some minor editing:

- Figure 4: If possible, please enhance the overall resolution of the map. Importantly, the legend shows some unnecessary grey edges, which can be easily (and quickly) rectified with photoshop or an open source equivalent

-Figure 7 (a and b): If possible, please enhance the overall resolution of the images and the Greek text (possibly requires new scanning of the original). Importantly, the texts feature an unnecessary grey background, which can be easily (and quickly) rectified with photoshop or an open source equivalent

- Figure 9 (a-c): Same as above. Importantly, however, please enlarge the entire image at least to half the page size, if possible. The texts are not legible when printed and when zoomed in digitally, they are blurred

- Graph 1: Please enhance the resolution of the graph. The text is not legible (either in print or digitally)

Content and References

The methodological and theoretical background, the presentation of the case studies as well as the final conclusions are well-thought and presented, sufficiently referenced (only in two minor cases did I miss a reference: compare text). The paper is overall convincing and interesting.

Apart from some minor content-related remarks highlighted in text, one question does remain: The epigraphical evidence on the water management system in the Hauran case study is obviously very important for the arguments drawn, esp. concerning the bi-directional interdependencies of water management issues between the rural areas and urban centers such as Kanatha or major religious sites such as at Si. However, can the cited inscriptions be more or less dated? If so, are they contemporary or can temporal developments/ variations of the water management system be detected?

This not an absolutely central issue for the larger argument of the paper, but perhaps a brief note concerning the (non-)contemporaneity of the texts could be added.

Author Response

All responses can be found in the word document

Thanks to the reviewer!

Katharina Rieger

Reviewer 2 Report

This is an original and interesting paper, which encourages to rethink established categories on centrality, arid regions, and their importance. The chosen combined approach of landscape archaeology, social network analysis, small world phenomena, and the study of weak and strong ties is very promising. The regional and chronological framework are the two Graeco-Roman landscapes of Hauran and Marmarica. Land, and particularly the Special Issue edited by Papantoniou and Vionis, is an adequate place for this research. Nevertheless, this study has the potential to be a better and long-lasting article, notably with regard to data presentation and explanations of the research design.

The paper is definitely noteworthy, but the following problems have to be addressed:

1. One needs to be very cautious with the starting point that arid regions have been generalized as marginal regions (from l. 9 onwards), especially from a comparative perspective. The hyper-arid to arid coast of Peru, for example, is widely regarded as the core area or center of many pre-Columbian cultures / "civilizations".

2. A more thorough check of the English could improve the fluency of the text.

3. The entire text should be carefully checked for spelling and punctuation (for instance, l. 154, 265, 330, 342, 492, and 648).

4. Something seems to be wrong with the citation on l. 119.

5. As in Fig. 11, it would help the reader in terms of orientation if the maps of the article had inset maps and, in some cases, north arrows (Figs. 2, 3, 4, 8, 10, 14, and 15), at least for Figs. 2 and 3.

6. One of the principal problems of the article are the questions why are the two case studies compared with each other and what makes them comparable beyond aridity? Just to give a few reasons would be sufficient here, even if these reasons are of practical nature.

7. In relation to point 6: What about the chronology of the two case studies? Again from a comparative point of view, it would simplify matters if the period(s) in question were also indicated in years and already at the beginning of the article. Furthermore, a short explanation about the chronological data would help.

8. Fig. 6: The author of the image is missing.

9. The quality of some images, especially Graph 1, seems to be quite poor.

10. The other principal problem is a missing introduction to the data used in this article, for instance settlement data and literary sources, and for which approach (landscape, social network analysis, small world, ties) the respective data is relevant. I am not suggesting to rewrite the complete text. A breakdown in the form of a table could suffice.

Author Response

(The authors gave the same response as above.)

Reviewer 3 Report

The paper examines the term of un-central landscape in Roman period. For this reason, the study focuses on the larger area than on a site, using examples are from Southern Syria and Northern Libya, both situated in arid environment. The study shows a network of relations and a bilateral interaction between central and marginal areas, depending on water management, agricultural production and religion. The author uses these examples of geographical and environmental marginal areas (arid environment) in order to prove that this kind of marginality does not mean either un-central area or without potentials.

The introduction presents the theoretical context of the study and allows to the reader understanding the terminology. The main body of the paper consists to the presentation of the two arid areas. A large part is dedicated to argue that, despite the geographical and environmental marginality of Northern Libya and Southern Syria, they could not be always considered as un-central areas. The argumentation is convinced. In the conclusion, the author summarizes in points the main results of the discussion.

Remarks to be considered by the author:

1.      The reviewer is not a native English speaker but it would be nice if a native English speaker (preferable an archaeologist) could proof-read the language.

2.      In the introduction, the author exposes very well the central place theory and the different terms used in the article and the state of the art.  However, the questions and the aims of the paper merit a better development at this part.

3.      The author explains well the term of centrality and the factors to which is associated. The reviewer would advice a better development of the human-human-landscape interactions in lines 165-168.

4.     The reviewer encourage the author to use geoarchaeological or/and geomorphological studies, if they exist for these regions, in order to better argue on the water management, water supply and wadis’ dynamic in Roman period. Is there any evidence of changes on river dynamic and sediment transport during the Greaco-Roman period? Could they help to better understand changes on the local economic of marginal areas?

Figure 1: the reviewer recommends the author to put on the map, at least the names of the regions under study

Figure 2 and 3: the north in the map should be appeared.

Line 188:  cite sites, dates and bibliographical references for water management systems.

Line 196-198: same as above.

Line 249: explain the “dichotomic view”.

Line 265: delete “the”.

Line 300: explain which the soft factors are.

Line 307: explain with some words what the Granovetter’s model is and which the strong and weak ties are.

Line 314: delete “can be reached”.

Line 351: delete “where”.

Figure 4: better quality of the map. The names of the sites are not easy-to-read.

Figure 5: scale and orientation are missing.

Figure 9: a bigger figure in order to allow the reading of the inscriptions.

Line 451: change “und” to “and”. Cite a reference.

Graph 1: the quality of the graph is not very good.

Line 492: delete the second “figure 11”. A better quality of the figure should be provided.

Figure 12 and 13: orientation is missing

Line 609-611: this statement should be more developed.

Author Response

All responses can be found in the word document

Thanks to the reviewer!

Katharina Rieger

Round  2

Reviewer 2 Report

The paper is accepted in the present form.